# Fecal miRNA Profiling of Yorkshire Terrier Enteropathy

**DOI:** 10.3390/ijms26073385

**Published:** 2025-04-04

**Authors:** Dana Mashaal, Magdalena Putzer, Patricia Freund, Hadi Shabanloo, Barbara Pratscher, Georg Csukovich, Katrin Spirk, Alexandro Rodríguez-Rojas, Iwan A. Burgener

**Affiliations:** Department for Small Animal and Horses, Clinic for Small Animals, University of Veterinary Medicine, 1210 Vienna, Austria; danasaber20@gmail.com (D.M.); barbara.pratscher@vetmeduni.ac.at (B.P.);

**Keywords:** canine IBD, IBD, inflammatory bowel diseases, chronic enteropathy, biomarker, miRNA

## Abstract

MicroRNAs (miRNAs) are small non-coding RNAs involved in gene regulation and are potential biomarkers for several diseases, including canine enteropathies. While metabolite profiling and microbiome in canine enteropathies have been previously explored, data on miRNA expression remain limited. This study aimed to profile miRNA expression in Yorkshire Terrier canine enteropathy using Illumina sequencing and quantitative PCR (qPCR) to compare miRNA levels between sick and healthy dogs from fecal samples. Despite the hypothesis that disease-related alterations in miRNA levels would differentiate sick dogs from controls, no significant differences were observed between the groups in either sequencing or qPCR analyses. These findings suggest that miRNA profiles may not vary significantly in the context of Yorkshire Terrier enteropathy and indicate that other molecular or metabolomic markers may be more indicative of disease state. This study also indicates that fecal samples may not be an ideal sample type for miRNA profiling. This study contributes to the understanding of molecular signatures in canine enteropathies and provides a basis for further research into alternative biomarkers for diagnosis and monitoring.

## 1. Introduction

Inflammatory bowel disease (IBD) is a chronic inflammation of the gastrointestinal tract that encompasses various conditions, including Crohn’s disease and Ulcerative Colitis in humans. Ulcerative Colitis causes explicit inflammation in the large intestine, whereas Crohn’s disease can affect any part of the gastrointestinal tract. Common symptoms include diarrhea, rectal bleeding, abdominal pain, weight loss, and fatigue. The exact cause of IBD is unknown, but it is believed to arise from a combination of genetic predisposition, environmental factors, and dysregulation of the gut microbiome, which triggers inappropriate immune responses [1].

Oxidative stress caused by an imbalance in Reactive Oxygen Species (ROS) production by the immune system is another contributing factor. This imbalance reduces the population of anaerobic bacteria while favoring aerobic bacteria, perpetuating chronic inflammation through a positive feedback loop involving macrophages. This cycle complicates efforts to resolve intestinal inflammation, exacerbating the disease [2].

Dysbiosis, a disruption in the composition and function of the gut microbiota, is another significant factor in IBD. Antibiotic exposure and other triggers can disturb the microbiome, initiating immune responses contributing to the disease. However, various strategies can restore a healthy microbiome, including dietary modifications (e.g., maintaining a diverse diet to minimize toxin exposure), prebiotics and probiotics, and fecal microbiota transplantation [3].

Several risk factors, such as smoking, certain medications, diet, autoantibodies, and allergies, are associated with IBD. Given the distinct histological differences between Crohn’s disease and Ulcerative Colitis, diagnosis is often specific to the type of IBD. For example, colonoscopy is commonly used to diagnose Ulcerative Colitis, while endoscopy is preferred for Crohn’s disease. Diagnostic tools include biopsies, imaging studies, stool sample analysis, and blood tests [4].

Treatment depends on the type and severity of IBD, as well as the patient’s response and tolerance to therapies. Corticosteroids are frequently used to suppress inflammation, while aminosalicylates, such as mesalazine, act locally in the gastrointestinal tract. Immunomodulators like methotrexate, which suppress the immune system and reduce inflammation, are also part of the treatment arsenal [5].

IBD in dogs presents similar conditions to those in humans, making dogs valuable models for advancing the understanding of this disease. Dogs and humans share environments, diets, and numerous microbiome species, contributing to these similarities. Despite this, there are differences between IBD in humans and dogs. Genetics and environmental factors are implicated in both species, but much about the disease remains poorly understood.

The small intestines of dogs with IBD differ from those of healthy dogs, suggesting a potential connection between the microbiome and IBD, like in humans. This highlights the link between gut bacteria and IBD in both species. Dogs are often treated with similar antibiotics and drugs used for humans [6]. Treatment depends on the severity of the disease and the dog’s response to specific medications. Dogs, particularly those experiencing food-related diarrhea, may significantly improve or even complete recovery from IBD through dietary changes, such as adopting a special diet with hydrolyzed protein. This shared understanding of IBD in dogs enhances their care and advances human comprehension of the disease.

More specifically, the manifestation of IBD in Yorkshire Terrier dogs, according to some retrospective studies, shows an enteropathy distinct from those observed in other breeds, referred to as “Yorkshire Terrier enteropathy” (YTE). This condition is characterized not only by classic chronic gastrointestinal (GI) symptoms but also by additional clinical manifestations such as low oncotic pressure and effusions. Laboratory findings commonly associated with YTE include hypoalbuminemia, hypocalcemia, and hypomagnesemia, which appear more frequently in affected individuals. Histopathological analysis often reveals hallmark features of YTE, including severe intestinal lymphatic dilation, crypt lesions, and villous atrophy.

We have recently studied a cohort of dogs affected by YTE, including through microbiome analysis [7] and metabolomics [8], with both studies showing alterations compatible with IBD such as dysbiosis and metabolic imbalance that do not revert during remission, at least in the short term. Because this cohort is very well studied and the diagnostic is confirmed, we have used selected individuals to also profile the micro-RNA (miRNA) content in fecal samples.

miRNAs are small non-coding RNAs, approximately 22 nucleotides long, that regulate gene expression by repressing or degrading their target mRNAs. They function in RNA silencing and post-transcriptional regulation, influencing protein synthesis. This mechanism involves miRNAs binding to the 3′ untranslated regions (UTRs) of target mRNAs, leading to mRNA degradation or translational repression. Biologically, miRNAs play essential roles in cell cycle regulation, development, apoptosis, differentiation, and immune response. Their involvement in gene regulation makes them pivotal in various diseases, particularly inflammatory pathways [9,10].

The versatility of miRNAs in regulating genes and biological processes has diverse applications. miRNAs are used as diagnostic tools through miRNA profiling and show promise as therapeutic targets by inhibiting their target mRNAs. Additionally, miRNAs function as biomarkers for disease diagnosis and prognosis, as their altered expression profiles across tissues can indicate disease states. Their tissue-specific expression profiles, determined by unique sequences, contribute to the specialization of biological functions [11].

In the IBD context, distinct miRNA expression patterns have been identified, highlighting their potential as biomarkers and therapeutic targets. Investigating the targets and pathways associated with specific miRNAs provides valuable insights into their functions and possible therapeutic interventions. For example, miRNAs such as miR-7-5p, miR-143-3p, miR-146a-5p, and miR-125b-5p exhibit anti-inflammatory properties by regulating cytokine expression or targeting genes in pro-inflammatory pathways. miRNAs like hsa-miR-29b-3p and hsa-miR-29c-3p contribute to tissue remodeling in IBD by modulating the extracellular matrix and fibrosis. Conversely, miRNAs such as miR-103a-3p and miRNA-122-5p inhibit cell proliferation and induce apoptosis [12].

miRNAs may act as oncogenes or tumor suppressors, depending on the context. For example, miR-215, miR-182-5p, and miR-320a-3p function as oncogenic miRNAs, while miR-143-3p, miR-375-3p, and miR-148b-3p act as tumor suppressors in various cancers [13,14,15,16,17,18,19,20]. miRNAs such as miR-125b and miR-223 are associated with Ulcerative Colitis (UC), showing upregulation in the colonic mucosa of UC patients. These miRNAs regulate the NF-κB pathway by modulating the expression of pro-inflammatory cytokines through TRAF6-mediated MAPKs and NF-κB signaling [21,22].

miRNAs also serve as diagnostic markers due to their differential expression in bodily fluids such as saliva and urine during inflammation, measurable through qPCR. Their specific expression across tissues, disease types, and stages makes them valuable tools for disease prognosis. Understanding the tissue-specific roles of miRNAs in IBD aids in uncovering disease pathogenesis, diagnosis, and treatment [23]. In this study, we hypothesized that miRNAs are differentially expressed in IBD compared to control conditions, in fecal samples, due to altered miRNA levels in disease states. To test this, we used a panel of miRNAs identified through miRNA sequencing from a cohort of healthy and IBD-affected dogs, quantified using qPCR.

## 2. Results

The animals sampled for this study are a sub-cohort of animals from previous studies and, hence, well characterized and with confirmed diagnostic [7,8,24]. In Table 1, a summary of the patients is presented.

Initially, fecal samples were collected, and their miRNA content was extracted and sequenced using the Illumina platform. Across the entire cohort, a total of 37 miRNAs were identified. However, precise quantification proved challenging, likely due to the low abundance of miRNAs in many of the samples. This low abundance may have limited the detection sensitivity and could be attributed to variability in sample quality or inherent differences in miRNA expression levels within the fecal material.

The identified miRNAs are summarized in Table 2, which provides an overview of their presence across the cohort. Despite the challenges in accurate quantification, these findings provide a valuable foundation for further experiments like the qPCR of fecal miRNAs to validate and to be able to quantify these miRNAs. Future studies may benefit from employing methods to enhance miRNA yield, such as optimized extraction protocols or amplification strategies, to improve the reliability of quantification in similar analyses.

After analyzing the qPCR datasets, we found no statistically significant differences in the relative expression of any analyzed miRNAs between dogs with YTE and healthy controls (Figure 1). This outcome suggests that the assessed miRNAs may not be differentially expressed in the context of YTE or that any potential expression changes may be too subtle to be detected with the methods and sample size used in this study.

Another consideration is the quality of the collected samples. Variations in sample integrity, RNA quality, or extraction efficiency could impact the detection and quantification of miRNAs, even when using robust and validated methodologies. Furthermore, miRNAs are known to exhibit dynamic and context-dependent expression patterns. It is possible that changes in miRNA levels occur at time points or under conditions not captured by the current study design.

These findings indicate the complexity of miRNA regulation in diseases such as YTE. While no significant differences were observed in this study, it highlights the need for further investigation with larger sample sizes, additional miRNA targets, and complementary approaches such as next-generation sequencing. Exploring other biofluids or tissues could also provide insights into differentially expressed miRNAs that were not detectable in the present sample set.

## 3. Discussion

MicroRNAs (miRNAs), as critical regulators of gene expression in diverse biological pathways, play an essential role in inflammation. In the context of inflammatory bowel disease (IBD)—a condition with a multifactorial and uncertain pathogenesis involving genetic predisposition, environmental triggers, and immune dysregulation—the evaluation of miRNAs has emerged as a promising approach for understanding disease mechanisms. Furthermore, miRNAs hold potential as clinical biomarkers for IBD diagnosis and therapeutic monitoring due to their differential expression profiles in the disease.

One of the unique properties of miRNAs is their stability against degradation, which enhances their potential as biomarkers, including in fecal samples. In this study, miRNA expression levels were assessed using quantitative PCR (qPCR) to explore their utility in distinguishing IBD-affected dogs from healthy controls.

Several technical challenges may have contributed to these results. High Ct values, which indicate a low abundance of target nucleic acid, raised concerns about genomic DNA contamination, suboptimal RNA isolation, or cDNA degradation. These issues are compounded by the complexities of stool samples, which are prone to contamination from food residues, bacterial RNA/DNA, mucus, and the microbiome.

In contrast to our study, prior research has demonstrated significant alterations in miRNA expression in IBD. For example, one study reported elevated levels of miR-16, miR-21, and miR-223 in blood and fecal samples from human IBD patients, particularly in those with active disease. Serum analysis revealed fold changes raging between 3 and almost 4 times for miR-16, miR-21, and miR-223. Similarly, fecal miRNAs were significantly upregulated, with miR-223 showing the highest expression (6.5-fold increase). These differences were more pronounced in Ulcerative Colitis (UC) compared to Crohn’s disease, reflecting distinct miRNA profiles in IBD subtypes. The stability of miRNAs in biofluids such as blood, urine, and faces, combined with their tissue-specific expression, underscores their potential as disease-specific biomarkers.

Studies in canine models partially support these findings. For instance, research has identified increased serum levels of miR-16, miR-21, miR-122, miR-146a, and miR-147, alongside decreased levels of miR-185, miR-192, and miR-223, in IBD-affected dogs compared to healthy controls. Additionally, tissue and blood analyses have revealed the upregulation of several miRNAs, including miR-21, miR-146a, miR-155, and miR-223, in IBD. However, these previous studies did not use directly sourced stool material as in our study.

The current study’s inability to detect significant differences in miRNA expression between IBD and healthy dogs may be attributed to the low disease severity in the cohort and technical limitations associated with stool sample complexity. Using fecal material as a source for microRNA biomarkers presents several challenges. The complex nature of feces, with its high microbial content and variable composition influenced by diet and gut health, can interfere with the extraction and accurate quantification of microRNAs. Additionally, sample degradation and inhibitors complicate standardization and reproducibility, making it difficult to interpret results reliably across different studies.

It is important to note that the experimental subjects in this study are patient dogs that live in various uncontrolled environmental conditions. While this variability reflects real-world scenarios and enhances the translational relevance of our findings, it also introduces potential confounding factors that may affect the interpretation of the results. As with many biomedical studies involving patient populations, these inherent variations represent a limitation that should be considered when evaluating this study’s outcomes.

Nevertheless, the broader body of evidence supports the utility of serum or plasma-based miRNA analysis as a promising approach for IBD diagnosis and monitoring. Future research should focus on refining sample processing techniques, incorporating larger and more diverse cohorts, and investigating the clinical relevance of miRNA expression in varying disease states to enhance diagnostic and prognostic capabilities.

In conclusion, while stool-derived miRNA analysis posed challenges in this study, according to the literature, serum and plasma remain more viable mediums for detecting miRNA biomarkers. Their potential to facilitate accurate and non-invasive diagnosis of IBD represents a significant step forward in advancing personalized medicine for inflammatory diseases.

## 4. Materials and Methods

### 4.1. Cases

All the samples used for this work were surplus materials from previous studies. The earlier studies were approved by the Ethics Committee of the University of Veterinary Medicine Vienna and the Austrian Federal Ministry of Science and Research (BMWF-68.205/0150-V/3b/2018), and the methods for sample collection were carried out following relevant Austrian guidelines and regulations.

Client-owned Yorkshire Terriers (*n* = 5) diagnosed with Yorkshire Terrier enteropathy (YTE) were prospectively enrolled from the Small Animal Internal Medicine Clinic of the University of Veterinary Medicine Vienna, Austria, between January 2019 and June 2021. as reported in previous studies [7,8,24]. The inclusion criteria included a history of chronic (≥3 weeks) or intermittent gastrointestinal (GI) signs, such as vomiting, diarrhea, anorexia, or weight loss, or evidence of pleural or abdominal effusion. A thorough diagnostic evaluation was performed to rule out other possible causes of GI signs or effusions. This evaluation included a physical examination, complete blood count (CBC), serum biochemical profile, measurement of bile acids and basal cortisol concentrations, ACTH-stimulation testing (if basal cortisol < 2 µg/dL), and the assessment of serum cTLI (canine trypsin-like immunoreactivity), SpecPL (specific pancreatic lipase), and cobalamin concentrations. Additional diagnostics included urinalysis for the urinary protein-to-creatinine (UPC) ratio, abdominal ultrasonography, fecal analysis by flotation and Giardia antigen testing, and gastroduodenoscopy at the time of presentation. Healthy adult Yorkshire Terriers without GI signs (*n* = 5) were prospectively enrolled as controls during the same period. Control dogs were determined to be healthy based on history, physical examination, CBC, serum chemistry profile, fecal analysis, urinalysis, and abdominal ultrasound. Exclusion criteria included previous antibiotic or glucocorticoid treatment within the last 2 weeks or feeding of a GI diet within the last 2 months.

### 4.2. Sampling

Fresh fecal samples were collected from all YTE dogs at the time of diagnosis and from healthy control dogs. Stool samples were transferred into sterile cryotubes using sterile plastic spoons and stored at −80 °C until RNA extraction.

### 4.3. RNA and miRNA Isolation

Total RNA, including miRNA, was isolated from fecal samples using the Qiagen miRNeasy Mini Kit according to the manufacturer’s protocol. RNA was stored at −80 °C until analysis. RNA concentration and purity were assessed using a X Qubit 2.0 Fluorometer Thermo Fisher Scientific and the kit Qubit™ microRNA Assay (Thermo Scientific, Waltham, MA, USA).

### 4.4. Small RNA Sequencing Library Preparation and Analysis

Small RNAs were converted into barcoded complementary DNA (cDNA) libraries for Illumina single-end sequencing using a 75-cycle protocol on the NextSeq500 platform (Illumina Inc., San Diego, CA, USA), following previously established methodologies [26]. The small RNA-seq analyses were conducted as described elsewhere [27], leveraging a curated miRNA reference database based on miRBase v23 and including the identification and characterization of novel miRNAs.

### 4.5. Reverse Transcription and cDNA Synthesis

Isolated miRNA concentrations were standardized to 10 ng/μL using nuclease-free water. Reverse transcription was performed on these standardized samples to synthesize cDNA, following the manufacturer’s protocol.

### 4.6. miRNA Primer Preparation

miRNA primers were diluted to a working concentration of 220 μL using nuclease-free water and stored in aliquots at −20 °C to minimize degradation from freeze–thaw cycles.

### 4.7. Quantitative PCR (qPCR)

Quantitative PCR (qPCR) was conducted using the miRCURY LNA SYBR Green PCR Kit (Qiagen) following the manufacturer’s protocol. cDNA samples were diluted 1:60 by combining 1 μL of cDNA with 59 μL of nuclease-free water. Each reaction was performed in a final volume of 10.05 μL, consisting of 3 μL of cDNA and 7.05 μL of Master Mix. A total of 34 miRNA primer sets were analyzed, with four sets run per 96-well plate. Each reaction consisted of two technical replicas and five biological repetitions represented by each animal for each study group for both the target miRNAs and housekeeping genes (Table 3). qPCR reactions were conducted on a QuantStudio™ 3 system, adhering to the cycling conditions specified in the miScript System Handbook. Data analysis was performed using QuantStudio™ 3 software v1.5.1 to evaluate the relative expression of miRNAs.

## Figures and Tables

**Figure 1 ijms-26-03385-f001:**
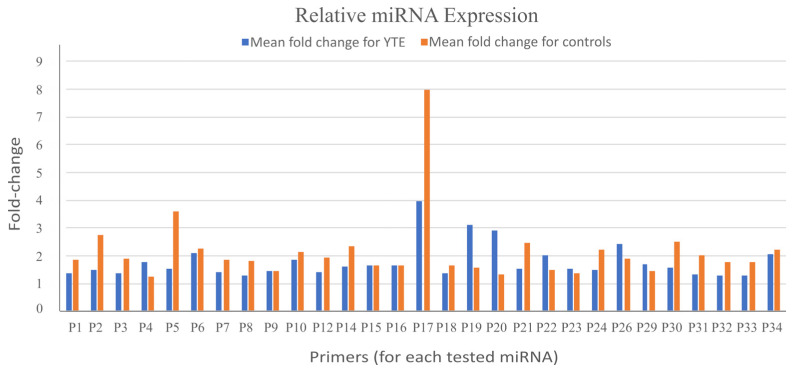
Normalized relative expression of target miRNAs using the comparative ΔΔCT method for primers P1–P34. miRNA expression levels were quantified from fecal samples collected from Yorkshire Terriers with and without enteropathy. Bars represent normalized data. No statistically significant differences were observed between sick and healthy dogs in either sequencing or qPCR analyses. The primers employed in this analysis were validated by the manufacturer (miRCURY™ microRNA, Qiagen [25]), ensuring high specificity and sensitivity for the selected miRNAs. Despite this, the absence of detectable differences could be attributed to several factors. One possibility is biological variability within the groups, where individual differences in miRNA expression may have masked subtle changes. Additionally, it is possible that the selected miRNAs are not directly involved in YTE pathophysiology, and the disease may instead involve other molecular pathways or regulatory mechanisms.

**Table 1 ijms-26-03385-t001:** The cohort characteristics of the patients included in this study. Sex is denoted as ‘F’ for females and ‘M’ for males. Age is represented as ‘Y’ for years and ‘M’ for months.

Dog ID	Breed	Age	Sex	Health Status
1	Yorkshire Terrier	12 Y 2 M	f	healthy
2	Yorkshire Terrier	7 Y 2 M	m	healthy
3	Yorkshire Terrier	12 Y 4 M	f	healthy
4	Yorkshire Terrier	6 Y 5 M	f	healthy
5	Yorkshire Terrier	6 Y 9 M	m	healthy
6	Yorkshire Terrier	8 Y 9 M	f	YTE
7	Yorkshire Terrier	7 Y 5 M	f	YTE
8	Yorkshire Terrier	6 Y 2 M	f	YTE
9	Yorkshire Terrier	7 Y 10 M	f	YTE
10	Yorkshire Terrier	7 Y 5 M	m	YTE

**Table 2 ijms-26-03385-t002:** A summary of miRNA identified from fecal samples via Illumina sequencing.

miRNA	Control 1	Control 2	Control 3	Control 4	Control 5	YTE1	YTE2	YTE3	YTE4	YTE5
miR-125-5p	0	19	0	0	0	0	0	0	0	0
miR-133	0	15	0	0	0	0	0	0	0	0
miR-7	22	13	0	0	0	29	13	0	0	17
miR-122-5p	0	0	0	80	0	0	12	0	0	0
miR-143-3p	0	0	0	45	0	0	0	0	0	0
miR-148a-3p	51	19	0	15	12	112	42	13	15	90
miR-148b-3p	11	0	0	0	0	16	0	0	0	13
miR-184-3p	0	38	13	38	0	22	0	10	0	0
miR-194-5p	155	79	0	50	21	191	104	27	10	238
miR-21-5p	164	41	36	349	19	149	76	23	56	272
miR-429-3p	0	0	0	0	0	16	0	0	0	20
miR-103	13	0	0	0	0	20	0	0	0	14
miR-16	113	58	11	52	21	271	101	22	23	272
miR-196	0	0	0	14	0	0	0	0	0	0
miR-22	15	0	0	0	0	11	0	0	0	12
miR-34a	0	0	0	0	0	15	0	0	0	0
miR-25	0	0	0	0	0	17	0	0	0	19
miR-29b	13	0	0	0	0	15	0	0	0	19
miR-29c	24	0	0	24	0	22	13	0	0	29
miR-182	0	0	17	0	0	0	0	15	0	0
miR-200c	49	18	0	12	10	113	34	0	0	81
miR-206	0	0	0	0	11	0	0	0	0	0
miR-26a	0	0	0	10	0	12	0	0	0	13
miR-26b	20	13	0	0	0	33	21	0	0	50
miR-27b	17	0	0	24	0	41	24	0	0	70
miR-320a	17	0	0	0	0	0	0	0	0	0
miR-378	18	11	0	0	0	24	18	0	0	43
miR-423-5p	15	0	0	0	0	0	0	0	0	0
miR-146a	13	0	0	0	0	19	10	0	0	12
miR-192	428	341	44	229	63	683	318	82	26	1428
miR-203	73	27	0	14	11	162	84	16	0	172
miR-215	22	12	0	0	0	26	0	0	0	26
miR-28	32	0	0	0	0	30	16	0	0	34
miR-29a	74	49	0	38	0	144	68	17	0	130
miR-375	25	14	0	0	0	18	13	0	0	22
miR-8859a	20	0	0	0	0	12	12	0	0	19
miR-93	14	0	0	11	0	0	0	0	0	18

**Table 3 ijms-26-03385-t003:** miRNA primers and housekeeping genes from miRCURY LNA miRNA Probe catalogue (Qiagen, Hilden, Germany). The primers were assigned random numbers for simplification purposes.

Number	Primer	Gene Globe ID (Qiagen)
1	hsa-miR-148b-3p	YP00204047
2	cfa-miR-8859a	YP02121094
3	hsa-miR-25-3p	YI04100613
4	hsa-miR-375-3p	YP00204362
5	hsa-miR-196a-5p	YP00204386
6	hsa-miR-184	YP00204601
7	hsa-miR-22-3p	YP00204606
8	hsa-miR-146a-5p	YP00204688
9	hsa-miR-93-5p	YP00204715
10	mmu-miR-429-3p	YP00205068
11	hsa-miR-122-5p	YP00205664
12	hsa-miR-423-5p	YP00205624
13	hsa-miR-125b-5p	YP00205713
14	bta-miR-20b	YP00205943
15	cfa-miR-652	YP02104033
16	bta-miR-26b	YP00205953
17	dme-miR-133-3p	YP00205954
18	hsa-miR-103a-3p	YP00204063
19	hsa-miR-28-3p	YP00204119
20	hsa-miR-34a-5p	YP00204486
21	hsa-miR-29b-3p	YP00204679
22	hsa-miR-29c-3p	YP00204729
23	hsa-miR-26a-5p	YP00206023
24	hsa-miR-27b-3p	YP00205915
25	hsa-miR-451a	YP02119305
26	hsa-miR-7-5p	YP02119317
27	hsa-miR-206	YP00206073
28	hsa-miR-143-3p	YP00205992
29	hsa-miR-378a-3p	YP00205946
30	hsa-miR-320a-3p	YP00206042
31	bta-miR-27a-3p	YP00205971
32	rno-miR-223-3p	YP00205120
33	hsa-miR-182-5p	YP00206070
34	cfa-miR-215	YP02102239
Housekeeping genes	Function	
Spike-In 2 (UniSp2)	Isolation Control for qPCR	YP00203950
Spike-In 4 (UniSp4)	Control of Isolation	YP00203953
Spike-In 6 (UniSp6)	Control of cDNA Synthesis	YP00203954

## Data Availability

All data are freely available under request.

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
