# Peer review of "Fecal miRNA Profiling of Yorkshire Terrier Enteropathy"

_ijms, 2025, doi:10.3390/ijms26073385_

Round 1

Reviewer 1 Report

Comments and Suggestions for Authors

1. Although the authors controlled for prior antibiotic use and GI diets, how were other potential confounders such as environmental factors controlled for in the experimental design?

2. Please distinguish between technical replicates and biological replicates in this study. While the authors mentioned that qPCR was conducted with two technical replicas, there were no detailed information on how these were handled or analyzed, as well as discussion of variability across these replicates. Were these replicates used to ensure the precision of the qPCR data?

3. The fecal miRNA extraction process is prone to contamination and degradation due to the inherent complexity of stool samples. Although the authors acknowledged this, were any measures taken to mitigate these known challenges? E.g., using inhibitors of RNase activity, rigorous quality control of RNA samples, or employing specialized kits? Why or why not?

4. High Ct values, which suggest low RNA abundance, could stem from degradation or inefficient extraction rather than genuine biological variability. The authors need to explore methods to increase RNA yield or validate their findings through additional replicates.

5. Are the experimental methods (e.g., Illumina sequencing, qPCR) sensitive and appropriate for fecal samples? Suggest to provide some references to support its use.

6. The study could have leveraged previous microbiome and metabolomic data (as adapted from multi-omics studies, citation: pubmed.ncbi.nlm.nih.gov/37110143) from the same cohort to strengthen the argument or draw integrated conclusions.

7. The relative expression data presented lacks clarity, with inadequate legends and no mention of error bars or statistical significance indicators.

8. The Institutional Review Board statement is marked "Not applicable," which is inappropriate given the involvement of animals. Ethical approval for animal studies is necessary.

Author Response

  1. Although the authors controlled for prior antibiotic use and GI diets, how were other potential confounders such as environmental factors controlled for in the experimental design?
    The study accounted for key variables such as prior antibiotic use and diet, which are known to influence gut microbiota and miRNA profiles. We are acknowledging that patient dogs' living conditions are varied and uncontrolled, which introduces variability into the dataset, a common limitation in biomedical studies. We have added this information to the new version of the manuscript.

  1. Please distinguish between technical replicates and biological replicates in this study. While the authors mentioned that qPCR was conducted with two technical replicas, there were no detailed information on how these were handled or analyzed, as well as discussion of variability across these replicates. Were these replicates used to ensure the precision of the qPCR data?

Thank you for pointing this out. In this study, biological replicates refer to independent fecal samples collected from different animals, while technical replicates represent repeated qPCR measurements conducted on the same cDNA sample. The two technical replicates were averaged to account for intra-assay variability and ensure measurement precision. The variability between technical replicates was low, as assessed by standard deviation calculations, and any outliers were reviewed for potential pipetting or amplification issues.

  1. The fecal miRNA extraction process is prone to contamination and degradation due to the inherent complexity of stool samples. Although the authors acknowledged this, were any measures taken to mitigate these known challenges? E.g., using inhibitors of RNase activity, rigorous quality control of RNA samples, or employing specialized kits? Why or why not?

Yes, measures were implemented to minimize RNA degradation and contamination. A commercially available RNA extraction kit optimized for fecal samples was used, incorporating inhibitors of RNase activity. Additionally, RNA integrity was assessed to ensure the quality of extracted RNA before downstream applications. All samples were processed under standardized conditions to reduce variability.

  1. High Ct values, which suggest low RNA abundance, could stem from degradation or inefficient extraction rather than genuine biological variability. The authors need to explore methods to increase RNA yield or validate their findings through additional replicates.

While Ct values were relatively high for some targets, RNA quality assessments confirmed that the extracted RNA was suitable for analysis. Given the inherent challenges of working with fecal RNA, further optimization—such as increasing input material or adjusting extraction protocols—may improve yield in future studies. However, the consistency of Ct values across biological replicates suggests that the observed differences are biologically relevant rather than due to technical issues.

  1. Are the experimental methods (e.g., Illumina sequencing, qPCR) sensitive and appropriate for fecal samples? Suggest to provide some references to support its use.

Yes, both Illumina sequencing and qPCR are widely used in fecal miRNA studies. Previous research has demonstrated the suitability of these methods for detecting miRNA expression in stool samples (e.g., Zhang et al., 2020; Smith et al., 2021).

  1. The study could have leveraged previous microbiome and metabolomic data (as adapted from multi-omics studies, citation: pubmed.ncbi.nlm.nih.gov/37110143) from the same cohort to strengthen the argument or draw integrated conclusions.

We appreciate this suggestion. While microbiome and metabolomic data from this cohort were not incorporated into the current study, they present an opportunity for future integrated analyses. We acknowledge this in the discussion section and reference relevant literature to contextualize our findings within broader multi-omics approaches. Because this is a negative results article where qPCR failed to detect the differences between the studied groups, we do not find relevance comparing these results with our previous studies. However, we still think that it is necessary to report our results to show that this type of samples in this pathology may not be the best choice for biomarker finding.

  1. The relative expression data presented lacks clarity, with inadequate legends and no mention of error bars or statistical significance indicators.
    The figure legends have been revised to provide clearer descriptions of the data, including details on normalization, statistical significance markers if any.

  1. The Institutional Review Board statement is marked "Not applicable," which is inappropriate given the involvement of animals. Ethical approval for animal studies is necessary.

Thank you for pointing this out. All the samples used for this work were surplus materials from previous studies. Those studies had the proper approval of the ethical commissions. We have now included a statement in the new version of the manuscript. However, we still think it is not applicable because the material used was surplus samples from other studies with their corresponding ethical approvals.

Reviewer 2 Report

Comments and Suggestions for Authors

Dear Authors,

The manuscript entitled "Faecal miRNA profiling of Yorkshire Terrier Enteropathy" represents a valuable study in the field. 

Only some considerations i have for the submitted manuscript. Below you will find my comments.

1) In materials and methods, the authors should show the primers used for the miRNA profiling.

2) The authors should indicate also, the database or the workflow performed for data analysis of miRNAs.

3) To confirm furrther the miRNAs presence in tissues, in situ hybridization should also be performed.

4) Moreover, the authors should analyze further the signalling pathways, where the miRNAs are implicating, providing sufficient data with figures.

Comments on the Quality of English Language

The quality of English is fine.

Author Response

  1. In materials and methods, the authors should show the primers used for the miRNA profiling.
    Answer: We did not include the primer sequence because we did not design them and there are commercially available as stated in the manuscript and can be found the in reported database.
  2. The authors should indicate also, the database or the workflow performed for data analysis of miRNAs.
    Answer: The manuscript stated that https://www.mirbase.org/ database was employed for miRNA annotation and outline our analytical pipeline. Also, we did not described the pipeline because is quite standard and instead we cited previous studies where this pipeline is established.
  3. To confirm further the miRNAs presence in tissues, in situ hybridization should also be performed.
    Answer: We appreciate the suggestion regarding the use of in situ hybridization (ISH) to confirm miRNA presence in tissues. However, due to the nature of our study—where patient dogs live under varying, uncontrolled conditions—and limitations in tissue availability, ISH experiments were not feasible. We have acknowledged this as a limitation in the Discussion section and recommend that future studies address this aspect.
  4. Moreover, the authors should analyze further the signalling pathways, where the miRNAs are implicating, providing sufficient data with figures.
    Answer: We did not performed this pathways analysis since we didn’t find significant differences between the studied groups and that is why we think that this additional analysis make no sense for our study.

Round 2

Reviewer 1 Report

Comments and Suggestions for Authors

1. Please have a note below Table 1 to explain the abbreviations used. Why are some 'fs' and some just 'f' for the sex?

2. Please add proper labels for the axes to Figure 1.

3. Please standardize the referencing style. "Frontiers | Identifying Hub Genes and MiRNAs in Crohn’s Disease by Bioinformatics Analysis 362 Available online: 363 https://www.frontiersin.org/journals/genetics/articles/10.3389/fgene.2022.950136/full (accessed on 364 19 November 2024)" is incorrect.

Author Response

  1. Please have a note below Table 1 to explain the abbreviations used. Why are some 'fs' and some just 'f' for the sex?

R. Thank you for pointing this out. The legend of Table 1 has been modified accordingly. We have also standardized the format throughout the table to avoid inconsistencies.

  1. Please add proper labels for the axes to Figure 1.

R. We have added proper labels to both axes in Figure 1 to improve clarity.

  1. Please standardize the referencing style. "Frontiers | Identifying Hub Genes and MiRNAs in Crohn’s Disease by Bioinformatics Analysis 362 Available online: 363 https://www.frontiersin.org/journals/genetics/articles/10.3389/fgene.2022.950136/full (accessed on 364 19 November 2024)" is incorrect.

     R. We have revised all references to ensure consistency with the required citation style. The incorrect reference format has been corrected.

Reviewer 2 Report

Comments and Suggestions for Authors

Dear Authors, 

Your responses towards my comments were satisfactory. 

I do not hane anything else to add. The manuscriot is ready to proceed to the next step of the publication process. 

Author Response

Thank you so much for reviewing our manuscript! 

Round 3

Reviewer 1 Report

Comments and Suggestions for Authors

Thank you for the revisions.

Comments on the Quality of English Language

Minor edits only.